# Corticosterone-Mediated Physiological Stress Alters Liver, Kidney, and Breast Muscle Metabolomic Profiles in Chickens

**DOI:** 10.3390/ani11113056

**Published:** 2021-10-26

**Authors:** Catherine L. J. Brown, Sarah J. M. Zaytsoff, Tony Montina, G. Douglas Inglis

**Affiliations:** 1Lethbridge Research and Development Centre, Agriculture and Agri-Food Canada, Lethbridge, AB T1J 4B1, Canada; kate.brown@uleth.ca (C.L.J.B.); zaytsoff@ualberta.ca (S.J.M.Z.); 2Department of Biological Sciences, University of Lethbridge, Lethbridge, AB T1K 3M4, Canada; 3Department of Agricultural, Food and Nutritional Science, University of Alberta, Edmonton, AB T6G 2P5, Canada; 4Department of Chemistry and Biochemistry, University of Lethbridge, Lethbridge, AB T1K 3M4, Canada; 5Southern Alberta Genome Science Centre, University of Lethbridge, Lethbridge, AB T1K 3M4, Canada

**Keywords:** chickens, physiological stress, corticosterone, feathers, ^1^H-NMR, metabolomics

## Abstract

**Simple Summary:**

Corticosterone is the major stress hormone in birds and research has shown that an increase in corticosterone can have adverse effects on bird health (e.g., predisposition to disease) production performance metrics. However, it is not currently possible to monitor commercial flocks for stress before performance is affected. A popular model of chicken stress involves administering corticosterone to chickens though their drinking water. However, corticosterone is non-polar so it must first be dissolved in ethanol, which means that the chickens are also drinking ethanol. In this study, an untargeted nuclear magnetic resonance-based metabolomics approach was used to investigate the effects of this model of stress in chickens, as well as the effects corticosterone on the chicken kidney, liver, and breast muscle metabolomes. We hypothesized that physiological stress modulates the metabolome of liver, kidney, and breast muscle due to increases in catabolism and gluconeogenesis. The administration of corticosterone altered the chicken liver, kidney, and breast muscle metabolomes. However, the ethanol carrier affected the metabolome of all three tissues, which indicated that corticosterone should be administered in an alternate fashion in future metabolomics studies to remove the confounding effects of ethanol. Furthermore, future research should focus on relating metabolite changes in tissues to non-destructive markers like blood, feces, or feathers to develop new diagnostic tools to better monitor on-farm stress during production.

**Abstract:**

The impact of physiological stress on the metabolomes of liver, kidney, and breast muscle was investigated in chickens. To incite a stress response, birds were continuously administered corticosterone (CORT) in their drinking water at three doses (0, 10, and 30 mg L^−1^), and they were sampled 1, 5, and 12 days after the start of the CORT administration. To solubilize CORT, it was first dissolved in ethanol and then added to water. The administration of ethanol alone significantly altered branched chain amino acid metabolism in both the liver and the kidney, and amino acid and nitrogen metabolism in breast muscle. CORT significantly altered sugar and amino acid metabolism in all three tissues, but to a much greater degree than ethanol alone. In this regard, CORT administration significantly altered 11, 46, and 14 unique metabolites in liver, kidney, and breast muscle, respectively. Many of the metabolites that were affected by CORT administration, such as mannose and glucose, were previously linked to increases in glycosylation and gluconeogenesis in chickens under conditions of production stress. Moreover, several of these metabolites, such as dimethylglycine, galactose, and carnosine were also previously linked to reduced quality meat. In summary, the administration of CORT in chickens significantly modulated host metabolism. Moreover, results indicated that energy potentials are diverted from muscle anabolism to muscle catabolism and gluconeogenesis during periods of stress.

## 1. Introduction

Chickens are exposed to many stressors during production. These include transportation from the hatchery to the barn, handling by people, and exposure to temperature fluctuations. When chickens experience a stressor, it stimulates the hypothalamic-pituitary-adrenal (HPA) axis and causes behavioral and physiological changes that help the chickens to cope with the stressor [1]. The glucocorticoid, corticosterone (CORT), is the major stress hormone in birds and is produced upon HPA activation [1,2]. Previous research has shown that an increase in the concentration of circulating CORT can lead to an altered cellular immune response [3] and that physiological stress negatively impacts the chicken immune system [4,5], which can have deleterious effects on production performance. For example, exposure to stressors can have a negative impact on poultry production metrics such as weight gain and feed conversion [6]. However, the systemic effects of stress on bird physiology, including impacts on metabolic processes, has not been extensively investigated to date.

One way to investigate the effects of stressors on metabolic processes is through metabolomics. Metabolomics is the study of the small molecular weight compounds, or metabolites, present in a tissue or biofluid, and allows for the analysis of the metabolic responses of living systems to external stimuli, such as stressors. An understanding of this response can provide valuable insight into the biochemical pathways that are potentially dysregulated due to physiological stress. Metabolomics can be used to study both pathological and physiological states, as well as to identify biomarkers in biological fluids and tissues. The metabolomics research that has been conducted in chickens to date is often focused on the detection of drugs and drug metabolites such as anti-virals [7], antibiotics [8], pesticides [9], and banned feed additives such as animal growth promoters [10] and chicken bone meal [11] in chicken meat. In addition, there were chicken metabolomic studies that investigated biomarkers of spoiled meat [12], the effects of different cooking methods [13], the effects of diet [14], and the quality of the final meat [15,16,17] and cooked product [18].

Nuclear magnetic resonance spectroscopy (NMR) is an experimental technique used to characterize the metabolome of various biological tissues and fluids that is high throughput, non-destructive, and highly reproducible [19,20,21]. Recent research has explored the use of NMR-based metabolomics to examine chicken tissues for both model development [22] and to identify biomarkers of disease [23,24]. NMR-based metabolomics was also utilized to study the effects of stress in rat and human models, and focused largely on pre-natal maternal stress [25,26] or early life stress [27] and not lifetime stress or its effect on animal production. Previous work from our team has utilized a targeted approach to NMR based metabolomics analysis, in which only certain subsets of metabolites were analyzed, and combined this analysis with both genomics and histology to investigate the effect of physiological stress [28]. This study relied on the addition of ethanol to drinking water to administer CORT to the chickens as proposed in Post et al. [29]; however, it did not investigate the effects of this delivery method, nor did it comprehensively examine the metabolic effects of CORT administration in an untargeted manner whereby all metabolites present in the tissues were considered.

Understanding the effects of physiological stress on metabolism in a comprehensive and untargeted manner is an important step towards identifying biomarkers of stress. This may facilitate the development of tools to help producers identify when stress may be adversely affecting their flocks so that mitigations can be employed early to prevent production losses. In the current study, we applied an untargeted NMR-based metabolomics approach to investigate the effects of both the ethanol-based delivery method of CORT administration and physiological stress, through dosage effects of CORT, on the metabolome of chicken kidney, liver, and breast muscle. We hypothesized that physiological stress modulates the metabolome of liver, kidney, and breast muscle due to increases in catabolism and gluconeogenesis.

## 2. Materials and Methods

### 2.1. Ethics Statement

The study was carried out in strict accordance with the recommendations specified in the Canadian Council on Animal Care Guidelines. The project was reviewed and approved by the Lethbridge Research and Development Centre (LeRDC) Animal Care Committee (Animal Use Protocol Review #1526) before commencement of the research.

### 2.2. Experimental Design

The study was designed as a factorial experiment with four levels of stress treatment and three levels of time arranged as a completely randomized design. The four stress treatments were: (1) control birds provided untreated drinking water (*n* = 9 chicks); (2) ethanol control birds provided with 0.2% ethanol in drinking water (*n* = 9); (3) 10 mg L^−1^ CORT in drinking water (*n* = 9); and (4) 30 mg L^−1^ CORT in drinking water (*n* = 9). The dose and method of CORT administration are detailed for a previous study [28]. Samples were obtained at 1, 5, and 12 days after initiation of the ethanol or CORT. Each replicate included 12 birds, and replicates were conducted on three separate occasions to ensure independence.

### 2.3. Corticosterone Administration

CORT (Sigma Aldrich Inc., Oakville, ON, Canada) was dissolved in 2 mL of anhydrous ethanol and added to 1 L of drinking water. Water containing CORT was prepared fresh each day and added to animal cages twice daily.

### 2.4. Bird Husbandry

Thirty-six specific-pathogen-free white leghorn chickens were used in this study. Eggs were purchased from the Canadian Food Inspection Agency (Ottawa, ON, Canada), and upon arrival at Lethbridge (shipped by air) were incubated in a Brinsea Octagon 40 Advanced Digital Egg Incubator according to the manufacturer’s recommendations (Brinsea Products Inc., Titusville, FL, USA). In this regard, eggs were maintained at 37.5 °C and 45% humidity with hourly turning for the first 18 days of incubation. Thereafter, eggs were set flat for hatching and humidity was increased to 60%. All hatched chicks were acclimatized in a group within one large animal pen (1.1 m^2^) for 10 days and had access to a brooder (Brinsea Products Inc., Titusville, FL, USA). Birds had ad libitum access to a non-medicated starter diet (Hi-Pro Feeds, Lethbridge, AB, Canada) and water. Birds were maintained on a 12 h light: 12 h dark cycle. At 11 days-of-age, birds were randomly assigned to the four stress treatments and housed within individually ventilated cages (IVCs) (Techniplast, Montreal, QC, Canada). Each animal cage contained a companion bird to ensure no birds were socially isolated. Corticosterone treatment began when chicks reached 14-days-of-age and continued until the end of the experiment. The experiment was repeated twice with individual birds treated as replicates (i.e., three replicate birds per treatment and time).

### 2.5. Sample Collection

One bird per IVC was randomly sampled at each of the three sample times. Birds were anesthetized with isoflurane (5% isoflurane; 1 L O_2_ min^−1^) and euthanized by cervical dislocation under general anesthesia. The abdomen was opened with a ventral midline incision and the liver, a kidney, and a sample of breast muscle were aseptically removed and stored at −80 °C until processing. Liver, kidney, and breast muscle were examined because the metabolome of these tissues was previously shown to be significantly altered in response to stress in chickens [14,28].

### 2.6. Sample Preparation

Tissues from liver, kidney, and breast muscle were homogenized in 4 mL g^−1^ methanol and 1.6 mL g^−1^ deionized water. Tissues were homogenized with 6-mm-diameter steel beads for 5 min intervals using a Qiagen Tissue-Lyser at 50 Hz followed by 1 min of vortexing. This step was repeated two additional times to ensure complete tissue homogenization. To each sample, 2 mL g^−1^ chloroform was added and vortexed thoroughly. Next, 2 mL g^−1^ chloroform and 4 mL g^−1^ deionized water were added to each sample and vortexed until thoroughly mixed. Samples were then incubated at 4 °C for 15 min to allow for protein precipitation and then centrifuged at 1000× *g* for 15 min at 4 °C. Next, 700 μL of the supernatant was removed and left until evaporated. Samples were rehydrated in 480 μL of metabolomics buffer (0.125 M KH_2_PO_4_, 0.5 M K_2_HPO_4_, 0.00375 M NaN_3_, and 0.375 M KF; pH 7.4). A 120 μL aliquot of deuterium oxide containing 0.05% *v*/*v* trimethylsilylpropanoic acid (TSP) was added to each sample (final total volume of 600 μL); TSP was used as a chemical shift reference for ^1^H-NMR spectroscopy. The solution was vortexed and then centrifuged at 12,000× *g* for 5 min at 4 °C to pellet any particulate matter. Following centrifugation, a 550 μL aliquot was loaded into a 5 mm NMR tube and run on a 700 MHz Bruker Avance III HD spectrometer (Bruker, Milton, ON, Canada) for spectral collection.

### 2.7. Nuclear Magnetic Resonance Data Acquisition and Processing

Spectra were collected on a 700 MHz Bruker Avance III HD spectrometer (Bruker, Milton, ON, Canada). The Bruker 1-D NOESY gradient water suppression pulse sequence ‘noesygppr1d’ was used with 10 ms mixing time. Each sample was run for 512 scans to a total acquisition size of 128 k, a spectral window of 20.5 ppm, a transmitter offset of 4.7 ppm, and a recycle delay of 4 s. All measurements were recorded using a Bruker triple resonance TBO-Z probe. The Bruker automation program “pulsal” was used on each sample before data acquisition to guarantee that the 90-degree pulse was calibrated correctly, ensuring quantitative and comparable data across samples [26]. The spectra were zero filled to 256 k, automatically phased, baseline corrected, and line-broadened by 0.3 Hz [25]. Spectra were then exported to MATLAB (MathWorks, Natick, MA, USA) as ASCII files, and underwent dynamic adaptive binning [30], followed by manual inspection and correction. Spectral binning resulted in 439, 460, and 379 spectral bins for kidney, liver, and breast muscle, respectively. The dataset was then normalized to the total metabolome, excluding the region containing the water peak, and pareto scaled.

### 2.8. Statistical Analysis

Spectral bins were subjected to both a univariate paired *t*-test and a multivariate variable importance analysis based on random variable combination (VIAVC) [31] analysis in MATLAB (Math Works, Natick, MA, USA) to determine which metabolites were significantly altered between treatments. For the paired *t*-test each comparison was paired within each replicate, and samples from all three time points, (1, 5, and 12 days after CORT administration) were pooled to provide an adequate sample size. The univariate measures were calculated using a decision tree algorithm as described by Goodpaster et al. [32]. All *p*-values obtained from this analysis were Bonferroni-Holm corrected for multiple comparisons. The VIAVC algorithm uses both Partial Least Squares Discriminant Analysis (PLS-DA) and the area under the Receiver Operating Characteristics (ROC) curve to determine the best subset of bins for group classification while also accounting for the significance of the bins; thus the algorithm considers the synergistic effects of bins [31]. MATLAB was also used to calculate the percent difference of the bins between treatments. The R package, MetaboanalystR [33] was used to carry out the Principle Component Analysis (PCA) and Orthogonal Partial Least Squares Discriminant Analysis (OPLS-DA). Metabolites were identified using the Chenomx 8.2 NMR Suite (Chenomx Inc, Edmonton, AB, Canada) and the complete list of significant metabolites was used to carry out pathway topology analysis using the Metaboanalyst’s Metabolomics Pathway Analysis (MetPA) web-based tool. The Hypergeometric Test was used to calculate relative betweenness centrality and out degree centrality measures to determine the importance of the compounds identified in the samples [34]; the KEGG database [35] for chicken pathways was selected for this analysis.

## 3. Results

### 3.1. Ethanol Alters the Metabolome of Kidneys, Liver, and Breast Muscle

Paired *t*-test and VIAVC tests revealed the bins that were significantly altered (*p* ≤ 0.05) between the control and ethanol control treatments in kidney (18 bins by *t*-test, 9 bins by VIAVC, no bins in common between the two tests), liver (53 bins by *t*-test, 9 bins by VIAVC, 5 common bins), and breast (no bins by *t*-test, 9 bins by VIAVC, no common bins). Examination of bins that were observed to be significantly altered via the paired *t*-test and/or the VIAVC analysis was done using a PCA scores plot which showed little to no unsupervised group separation in any of the tissues (Appendix A). Subsequent OPLS-DA analyses for the ethanol control treatment showed supervised group separation and a good model fit (as indicated by the Q^2^) in liver (Q^2^ = 0.85, *p* < 0.001, R^2^ = 0.97, *p* < 0.050) and breast (Q^2^ = 0.66, *p* < 0.001, R^2^ = 0.76, *p* < 0.050), but not kidney (Q^2^ = 0.16, *p* = 0.076, R^2^ = 0.65, *p* = 0.062) tissues (Figure 1).

Double cross-validation and permutation tests validated the observed supervised separation results as a function of ethanol in the drinking water in liver and breast (*p* < 0.001), but not kidney (*p* = 0.076) tissues. Quantities of specific metabolites differed (*p* < 0.050) between the control and ethanol control treatments as determined by paired *t*-test and/or VIAVC analysis. In the kidney, 5/10 (50%), in liver, 11/27 (40%), and in breast muscle, 1/5 (20%) unique metabolites were significantly up-regulated (Figure 2). Pathway topology analysis showed that the inclusion of ethanol in the drinking water affected aminoacyl-tRNA biosynthesis (*p* < 0.001), and alanine, aspartate, and glutamate metabolism (*p* < 0.050) in kidney and liver, as well as D-glutamine and D-glutamate metabolism and nitrogen metabolism (*p* < 0.050) in kidney and breast muscle (Appendix A). Only in the kidney was the branched chain amino acid (BCAA) (e.g, valine, leucine and isoleucine) degradation pathway altered (*p* < 0.050). In the liver, other amino acid metabolism pathways that were altered included the phenylalanine metabolism pathway (*p* < 0.050) and the arginine and proline metabolism pathway (*p* < 0.050).

### 3.2. Corticosterone Substantively Alters the Metabolome of Kidneys, Liver, and Breast Muscle

To ascertain the impact of CORT, the metabolome of birds administered CORT at a dose of 10 mg L^−1^ or 30 mg L^−1^ was compared to birds administered ethanol without CORT in drinking water to eliminate metabolite bias as a result of the ethanol carrier. Paired *t*-test and VIAVC tests were applied to each CORT treatment. The liver tissues had the largest number of significantly altered bins across all tissues and dosages (Appendix A). The relative concentration changed for 46 metabolites in kidney, 71 in liver, and 75 in breast with CORT administration (Appendix A). PCA score plots show that at the lower dose of CORT there was no unsupervised separation in kidney tissue (Appendix A), but there was unsupervised separation in liver and breast muscle tissues (Appendix A) with some confidence interval overlap. At the higher dose of CORT there was unsupervised separation observed in all three tissues (Appendix A) with complete separation of the confidence intervals for breast muscle and less overlaps for both the kidney and liver tissues. Subsequent OPLS-DA analyses showed supervised group separation and a good model fit for 10 mg L^−1^ CORT in liver (Q^2^ = 0.80, *p* < 0.001, R^2^ = 0.99, *p* < 0.050) and breast muscle (Q^2^ = 0.44, *p* = 0.050, R^2^ = 0.63, *p* < 0.050), but not in kidney (Q^2^ = 0.02, *p* = 0.151, R^2^ = 0.66, *p* = 0.111) (Figure 3A,C,E). Likewise, supervised separation was observed for the 30 mg L^−1^ CORT dosage in liver (Q^2^ = 0.81, *p* < 0.001, R^2^ = 0.93, *p* < 0.001), breast muscle (Q^2^ = 0.87, *p* < 0.001, R^2^ = 0.92, *p* < 0.001), and kidney (Q^2^ = 0.71, *p* < 0.001, R^2^ = 0.92, *p* < 0.050) (Figure 3B,D,F). At 10 mg L^−1^, CORT significantly up-regulated 39/43 (90%) of the unique metabolites in the kidney, 20/115 (17%) in the liver, and 20/25 (80%) in breast muscle. At 30 mg L^−1^, CORT significantly up-regulated 13/36 (36%) of the unique metabolites in the kidney, 26/93 (28%) in the liver, and 17/96 (18%) in breast muscle (data not shown). There were 11 metabolites that were significantly altered at both the 10 and 30 mg L^−1^ CORT doses in kidney, 46 in liver, and 14 in breast muscle (Figure 4). Glucose and N-methylhydatoin were significantly altered in all three tissues. In kidney and breast muscle, all metabolites that were altered at both doses showed a greater percentage difference at 30 mg L^−1^ CORT than at 10 mg L^−1^ CORT. In liver there were 16 metabolites that showed a greater change in regulation at 10 mg L^−1^ CORT than at 30 mg L^−1^ CORT, and 27 metabolites that followed the same trend observed in kidney and breast muscle, with three metabolites that had equal changes in regulation at both concentrations of CORT.

Pathway topology analysis indicates that the inclusion of 10 mg L^−1^ CORT in the drinking water affected galactose (*p* < 0.001); starch and sucrose (*p* < 0.050); and inositol phosphate (*p* < 0.050) metabolism. In liver and kidney alanine, aspartate, and glutamate (*p* < 0.050); and histidine (*p* < 0.050) metabolism were altered (data not shown). For the 30 mg L^−1^ CORT, treatment pathway topology analysis indicated that two pathways were altered in all three tissues; these were the aminoacyl-tRNA biosynthesis (*p* < 0.001) and galactose metabolism (*p* < 0.050) pathways (Figure 5). In liver and kidney, glutathione (*p* < 0.050), and starch and sucrose (*p* < 0.050) metabolism were also altered. In liver and breast muscle, pantothenate and CoA biosynthesis (*p* < 0.050), histidine (*p* < 0.050), and purine (*p* < 0.050) metabolism were also all significantly altered. In kidney and breast muscle, alanine, aspartate, and glutamate metabolism (*p* < 0.050), pyruvate metabolism (*p* < 0.050), and glycine, serine, and threonine metabolism (*p* < 0.050) were also altered.

## 4. Discussion

Metabolomics provides a comprehensive understanding of all the responses of whole living systems to pathophysiological stimuli, genetic modification, diet, and the environment. Furthermore, metabolomics is particularly adapted to identify biomarkers as it provides a snapshot of all the metabolic activity occurring in a system [21,36]. For these reasons identifying biomarkers of physiological stress and disease using metabolomics is gaining more momentum in human biomedical research, as well as in poultry science [23,37,38].

When chickens are exposed to a stressor, the HPA axis is activated to help the bird cope with the stressor [3,39].This activation leads to an increased concentration of circulating CORT, the main glucocorticoid in birds. Factors that stimulate CORT release are relevant to poultry producers as the hormone can be linked to detrimental effects on bird health [23,40], growth rate [41], and the overall quality of the meat [17,42]. Previous studies showed that CORT decreases mass gain [41,43,44], increases the amount of lipids in the liver [43], decreases skeletal muscle growth [41], and decreases nitrogen retention [45] by increasing the degradation of muscle proteins [41,44]. In addition, CORT has previously been shown to cause an increase in plasma glucose [41] and fatty acid concentrations [45]. In the current study, we investigated the effect of CORT administered at two doses in drinking water (10 and 30 mg L^−1^) on white leghorn chickens using ^1^H-NMR metabolomics, and we observed distinct changes in the metabolomes of liver, kidney, and breast muscle regardless of the dose administered. Moreover, CORT administration was observed to alter several metabolite concentrations that can be linked to alterations in metabolic pathways related to both bird health and meat quality.

### 4.1. Ethanol Impacts on Kidneys

Previous studies reported that rats administered ethanol in drinking water exhibit reduced renal function due to kidney injury [46,47]. Both the liver and the kidney have the ability to catabolize ethanol, but the formation of free radicals occurs during catabolism, which can damage these organs and cause energy intensive inflammatory responses [48]. As a result of increased energy demand, the process of gluconeogenesis can be utilized by the kidney to convert glycerol into glucose and to provide energy for cells in need [49,50]. The decrease in glycerol concentrations observed in the kidney of birds administered ethanol supports this mechanism, and suggests that gluconeogenesis was increased in the kidney to provide more energy due to the higher energy demands due to the inflammatory response caused by the ethanol. In chickens, gluconeogenesis occurs in both the liver and the kidney; however, the kidney is the primary site of gluconeogenesis in chickens [51]. A further indication that the kidney indeed had higher energy demands is supported by the observed decrease in alanine. The Cahill cycle, or the glucose-alanine cycle, occurs in most extrahepatic tissues, and the resultant molecules from the breakdown of BCAA are converted to alanine, which can more easily be transported to the liver where it can be converted into glucose [50,51,52,53]. Although we did not observe an increase in alanine within the livers of birds administered ethanol, the accumulation of alanine could have been obscured by the rapid conversion of alanine to urea via the urea cycle within the liver.

### 4.2. Ethanol Impacts on Liver

The liver is the primary site of ethanol catabolism, and chronic ethanol ingestion and metabolism causes mild damage to the liver though oxidative stress [48] and can induce fatty liver disease [48,54]. Furthermore, the administration of ethanol to chickens has been shown in previous studies to decrease liver antioxidant enzyme activities [55]. The combination of increased oxidative stress and reduced antioxidant enzymes would be expected to result in an increase in energy demands in the liver. The liver is the major site of the Cori cycle in chickens, in which lactate is converted to glucose to supply these energy demands [45,51]. The observed down-regulation of lactate in the livers of the ethanol control birds in the current study is consistent with an increase in the use of energy in chickens [14]. We also observed that phenylalanine was down-regulated in the livers of birds administered ethanol. Phenylalanine is a precursor for tyrosine, which is then used for the synthesis of neurotransmitters [56], and a decrease in phenylalanine concentration may adversely affect a chicken’s ability to respond to stimuli [14].

### 4.3. Ethanol Impacts on Breast Muscle

In breast muscle of birds administered ethanol, the metabolites of the glutamine and glutamate metabolism pathways, along with the metabolites of the nitrogen metabolism pathway, were significantly altered. Glutamine is normally the most abundant amino acid in healthy skeletal muscle; however, when muscle is stressed, the concentration of glutamine drops significantly. Therefore, glutamine is designated as a conditionally essential amino acid during periods of stress [57,58]. Studies have shown the glutamine supplementation in broilers was associated with improved growth, carcass characteristics, and overall meat quality while under stress [57,59]. Under stress conditions, there is a depletion of glutamine [59], as was observed in breast muscle of birds administered ethanol in the current study, suggesting that ethanol can lead to a decrease in breast meat quality.

### 4.4. Corticosterone Impacts on the Kidney Metabolome

The concentration of glutamate in the kidney was decreased in CORT stressed birds regardless of the concentration administered. Glutamate is a neurotransmitter, but is also present in peripheral tissues, like kidney, where it plays an important role in energy production, nitrogen metabolism, and the body’s response to oxidative stress [50,60]. Glutamate receptors in the kidney are coupled to a number of G-protein cascades [60]. There are many N-methyl-D-aspartate (NMDA) receptors in the kidney to which glutamate can bind. Although glutamate spectra are included in the Chenomx database for identification, glutamate bound to its receptor is not, meaning that bound glutamate cannot be identified in ^1^H-NMR spectra. The lower concentration of glutamate that we observed in the kidney of birds administered CORT could be caused by greater amounts of the metabolite being bound to its receptor. The activation of these receptors affects renal function, and in some cases may induce renal dysfunction; for example, a previous study showed that prolonged exposure to NMDA caused excessive NMDA receptor activation that leads to an accumulation of reactive oxygen species [60] which are toxic for renal cells; and this agrees with the metabolite changes observed in the current study. In addition, the concentration of glucose in the kidneys of the CORT birds was increased, and this agrees with the previously suggested increased role of the Cahill cycle. In the Cahill cycle, alanine is converted to glucose, and one key source of alanine is the conversion of glutamate and pyruvate into alanine and alpha-ketoglutarate. Previous studies also showed an increase in glucose in kidneys under stressed conditions [50,51,52,53]. This suggests that the combination of relative changes in both glucose and glutamate might serve as a strong biomarker of stress in chickens.

All metabolites altered in the kidney following CORT administration were affected to a greater extent in birds administered CORT at the 30 mg L^−1^ dose relative to the 10 mg L^−1^ dose. For example, the concentration of 4-hydroxyproline was substantially increased for birds administered CORT at the higher dose. In most animals, including chickens, hydroxyproline is required for the synthesis of glycine. Glycine is known to scavenge oxidants and help regulate the redox state of cells [61]. Moreover, up-regulation of hydroxyproline has been observed in response to many different kinds of stress, such as inflammatory and genotoxic stresses [62,63]. Elevated concentrations of malate were linked to kidney dysfunction in rats [64], and the increased concentration of malate observed in the kidneys of the birds administered CORT could indicate that the CORT treatment is a biomarker of renal dysfunction and this warrants further investigation.

### 4.5. Corticosterone Impacts on the Liver Metabolome

A key function of the liver is to provide glucose to tissues. Under conditions of stress the liver increases rates of both glycogenolysis and gluconeogenesis to meet the body’s increased energy demands [65]. Amino acids represent the main metabolic source of energy for the liver [66,67]. The essential amino acids that were identified in the liver of CORT-treated chickens in the current study (i.e., threonine, valine, isoleucine, methionine, phenylalanine, and tryptophan) were all down-regulated, indicating that energy demands in the liver were higher in birds administered CORT (i.e., relative to birds administered ethanol alone). This agrees with previous research that showed that concentrations of amino acids drop in liver under stress due to increases in metabolic demands [65,67,68].

In contrast to essential amino acids, mannose concentrations were significantly increased in the livers of birds administered CORT at 10 and 30 mg L^−1^ as compared to the ethanol control treatments. Previous studies observed up-regulation of mannose-6-phosphase, which is indistinguishable from mannose in 1D ^1^H-NMR spectra; mannose-6-phosphate is an integral metabolite in the formation of GDP-mannose which is important for N-linked glycosylation [67]. Glycosylation has been shown to be increased under acute stress conditions [67], and mannose is often used as the substrate for glycosylation if the cell experiences stress [69]. The function of glycosylation under acute stress is poorly understood at present, but is likely involved in the regulation of heat shock proteins, which are often up-regulated under various conditions of stress [69].

Previous studies observed an increase in glucose in chicken livers when the birds were stressed by high temperatures [67]. The up-regulation of glucose observed in the livers of chickens administered CORT in the current study is consistent with these findings, thereby suggesting that increases in the concentration of glucose in the liver may be indicative of physiological stress regardless of the incitant. Furthermore, excessive CORT exposure was shown to cause fatty liver in chickens [70,71]. Steatosis was noted in the livers of the birds receiving CORT after 5 days, which was previously published on the same cohort of birds from which tissues were obtained and analyzed in the current study [28].

### 4.6. Corticosterone Impacts on the Breast Muscle Metabolome

We observed that BCAAs in breast muscle were significantly down-regulated in CORT-treated birds as compared to control treatment birds. BCAAs are mostly metabolized in skeletal muscle and other peripheral tissues [72], and the decrease in BCAAs observed in the current study is consistent with an increase in metabolic demands imparted due to stress. Protein synthesis is an energy intensive process, and much more so than proteolysis [73,74]. Thus, insufficient energy to build protein, as would be the case if the birds’ energy was used for other metabolic processes, may have contributed to lower body weights that were previously observed on this cohort of stressed chickens [28]. It is noteworthy that stress is known to reduce feed intake [67,75]. Although we did not measure feed intake in the current study, it is possible that the decrease in bird weights could be caused entirely or partly by a reduced appetite in the birds, as seen in previous studies [67]. However, reduced food intake would still be expected to result in the observed changes in BCAAs related to muscle metabolism and increased energy demand. It is widely accepted that high concentrations of circulating CORT suppress muscle growth and protein synthesis [65], and can even lead to muscle atrophy [76], which would also contribute to the lack of mass gain in the CORT chickens.

We observed an increased concentration of glucose in breast muscle of birds administered CORT as compared to control treatment birds. As indicated previously, stress increases gluconeogenesis in the liver [17,67]. This results in increased levels of glucose in liver tissue, as well as transport of glucose to other tissues [77]. Stress was shown to increase the level of glucose in breast muscle [28,78], and previous studies have correlated meat quality to glucose levels [16], with higher quality breast meat possessing lower levels of glucose than poorer quality meat [16,42]. Thus, the findings of higher concentrations of glucose in breast muscle of birds administered CORT suggests that stress contributes to poor meat quality. Consistent with this conclusion, anserine, carnosine, proline, dimethylglycine, and phenylalanine concentrations were significantly reduced, while mannose-6-phosphate was significantly increased in CORT-treated birds. This agrees with previous research that showed similar concentration changes in these metabolites associated with overall meat quality measures [16,38,79].

Galactose metabolism is a pathway that has previously been determined to be significantly up-regulated in lower quality breast meat samples [16]. Metabolites associated with the galactose metabolism pathway, including glucose and glucose-1-phosphate, were observed to be significantly up-regulated in birds administered CORT as compared to the control birds in the current study, further implicating stress as an important contributor to meat quality. Overall, we identified and confirmed a variety of breast muscle metabolites that were significantly altered due to physiological stress incited by CORT, which may be of value as biomarkers of stress and impacts of stress on breast meat quality.

## 5. Conclusions

In the current study we showed that ethanol altered the metabolomes of the liver, kidney, and breast muscle of chickens; however, the effects of ethanol alone were substantially less than the metabolomic effects of CORT orally administered in ethanol. It is possible that CORT and ethanol together exerted a larger effect than either alone would have. Study results indicate that CORT should be administered in an alternate fashion (e.g., incorporated into the feed) in future studies to remove the confounding effects of ethanol on the metabolome. Despite the observed effects of ethanol, the data from this study showed that the stress hormone, CORT, was linked to changes in metabolite concentrations across several important tissues in chickens that could be linked to overall health and production. This includes kidney glutamate concentrations, liver amino acid concentrations, and increases in glucose concentrations in the kidney, liver, and breast muscle of white leghorn chickens. Significantly, several stress induced changes in metabolite concentrations observed in breast muscle may be associated with meat quality which warrants further investigation. Moreover, alterations to the metabolome of chickens incited by CORT were consistent with increases in metabolism due to stress induction, which were consistent with a previously published reduction in the mass of CORT administered birds relative to control treatments in the same chicken cohort. Future research should focus on relating metabolite changes in tissues to non-destructive markers like blood, feces, or feathers relative to metrics of birds’ health and meat quality with the goal of developing new diagnostic tools to better monitor on-farm stress. Ultimately, the identification of quantitative biomarkers of stress will allow producers to utilize the markers to develop and objectively evaluate mitigation strategies to proactively enhance bird health and flock performance.

## Figures and Tables

**Figure 1 animals-11-03056-f001:**
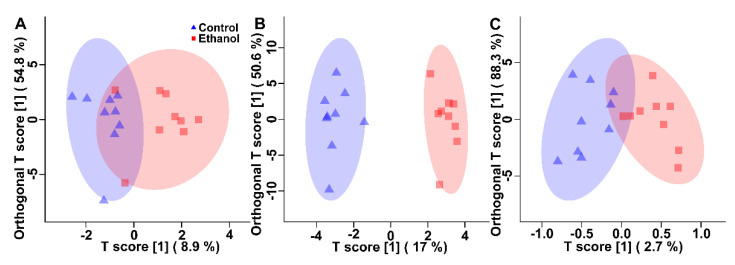
Orthogonal partial least squares determinant analysis score plots for the control treatment vs. the ethanol control treatment. (**A**) Kidney (Q^2^ = 0.16, *p* = 0.076, R^2^ = 0.65, *p* = 0.062). (**B**) Liver (Q^2^ = 0.85, *p* < 0.001, R^2^ = 0.97, *p* < 0.050). (**C**) Breast muscle (Q^2^ = 0.66, *p* < 0.001, R^2^ = 0.76, *p* < 0.050). Each triangle or square represents one chicken under study (*n* = 9), plotted using a list of bins found to be statistically significant via paired *t*-test and/or multivariate variable importance analysis based on random variable combination analysis. The x- and y-axis represent the predictive (between group separation) and orthogonal (within group variation) components of the data, respectively. The shaded ellipse represents the 95% confidence interval.

**Figure 2 animals-11-03056-f002:**
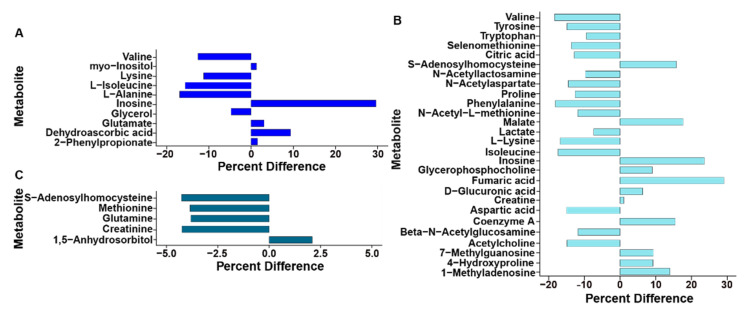
Percent difference for metabolites significantly altered in the control treatment when compared to the ethanol treatment. (**A**) Kidney. (**B**) Liver. (**C**) Breast muscle. All metabolites were significantly altered based on paired *t*-test and/or multivariate variable importance analysis based on random variable combination analysis.

**Figure 3 animals-11-03056-f003:**
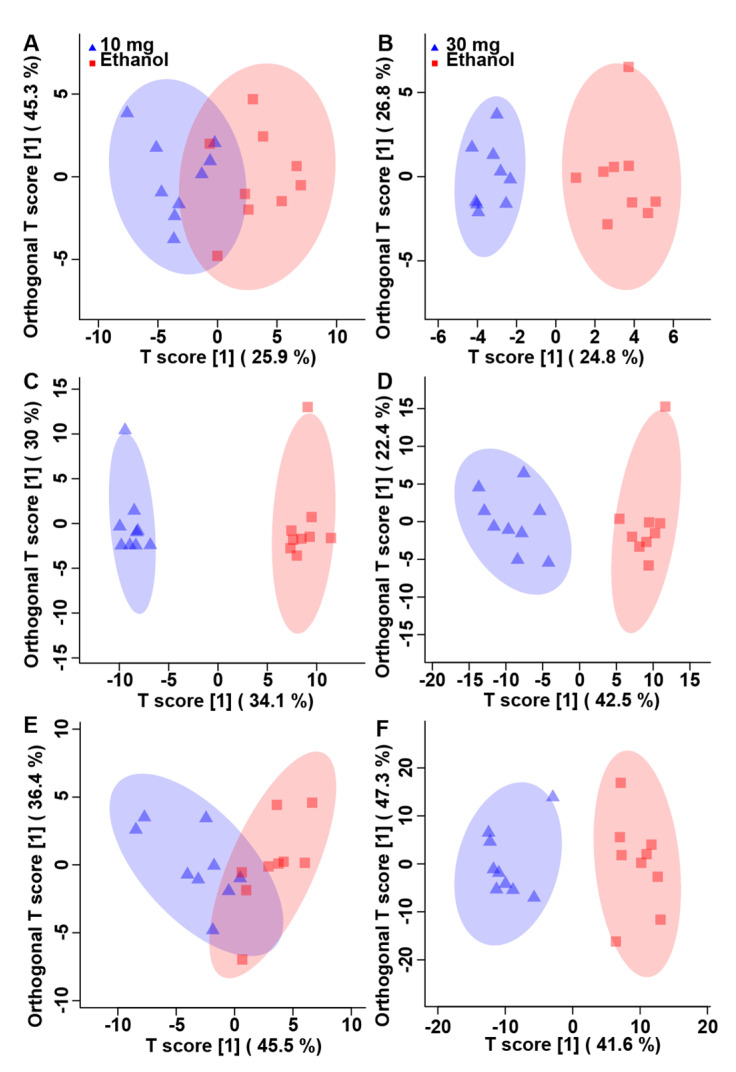
Orthogonal partial least squares determinant analysis score plots for the corticosterone treatments vs. the ethanol control treatment. (**A**) Kidney—10 mg L^−1^ corticosterone (Q^2^ = 0.02, *p* = 0.151, R^2^ = 0.66, *p* = 0.111). (**B**) Kidney—30 mg L^−1^ corticosterone (Q^2^ = 0.71, *p* < 0.001, R^2^ = 0.92, *p* < 0.050). (**C**) Liver—10 mg L^−1^ CORT (Q^2^ = 0.80, *p* < 0.001, R^2^ = 0.99, *p* < 0.050). (**D**) Liver—30 mg L^−1^ corticosterone (Q^2^ = 0.81, *p* < 0.001, R^2^ = 0.93, *p* < 0.001). (**E**) Breast muscle—10 mg L^−1^ corticosterone (Q^2^ = 0.44, *p* = 0.050, R^2^ = 0.63, *p* < 0.050). (**F**) Breast muscle—30 mg L^−1^ corticosterone (Q^2^ = 0.87, *p* < 0.001, R^2^ = 0.92, *p* < 0.001). Each triangle or square represents one chicken under study (*n* = 9), plotted using a list of bins found to be statistically significant via paired *t*-test and/or multivariate variable importance analysis based on random variable combination analysis. All three tissues show supervised separation at the highest corticosterone dose. All but kidney also show supervised separation at the lower dose. The x- and y-axis represent the predictive (between group separation) and orthogonal (within group variation) component of the data, respectively. The shaded ellipse represents the 95% confidence interval.

**Figure 4 animals-11-03056-f004:**
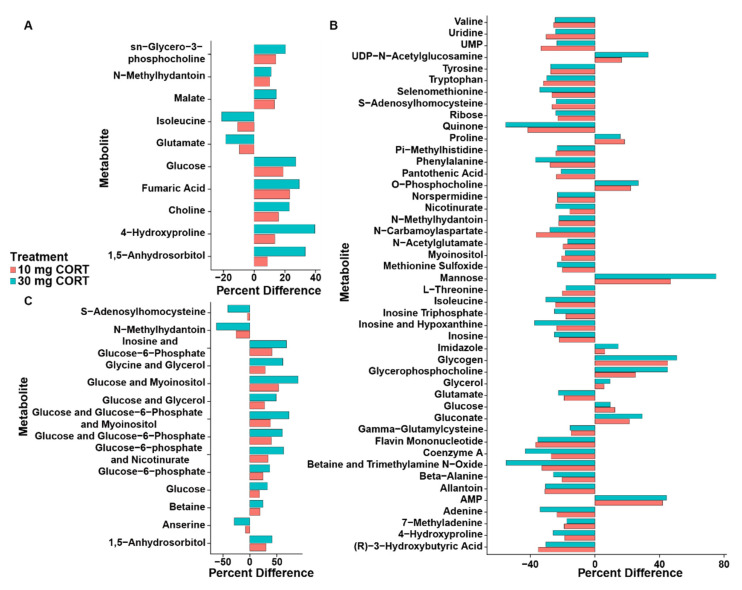
Metabolites significantly altered in both the 10 and 30 mg L^−1^ corticosterone treatments. (**A**) Kidney. (**B**) Liver. (**C**) Breast muscle. All metabolites were significantly altered (*p* < 0.050) as compared to the ethanol control treatment based on paired *t*-test and/or multivariate variable importance analysis based on random variable combination analysis.

**Figure 5 animals-11-03056-f005:**
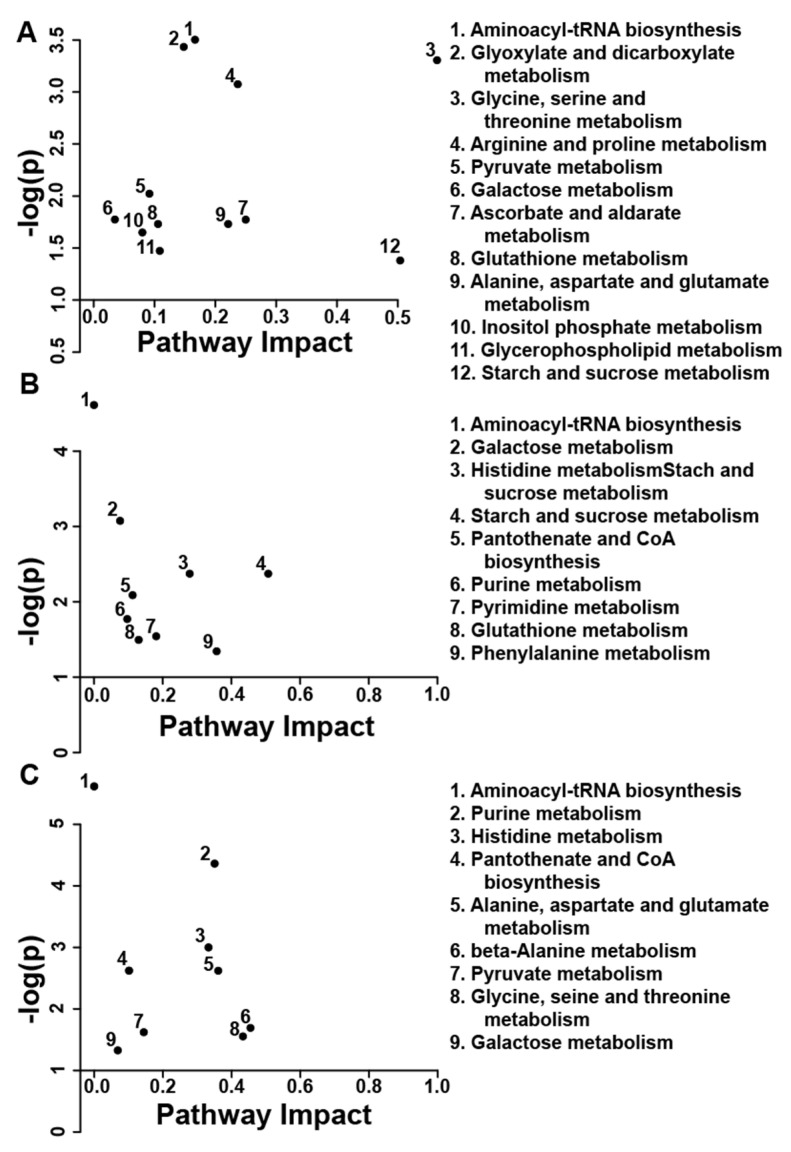
Metabolomic pathway topology analysis showing all matched pathways according to *p*-values from pathway enrichment analysis and pathway impact values for corticosterone administered at a dose of 30 mg L^−1^ vs. the ethanol control treatment. (**A**) Kidney. (**B**) Liver. (**C**) Breast. A larger value on the *y*-axis indicates a lower *p*-value. The *x*-axis gives the pathway impact as calculated using the hypergeometric test. All metabolic pathways with *p* < 0.050 are shown. This figure was created using the lists of metabolites identified as significant by paired *t*-test or multivariate variable importance analysis based on random variable combination testing.

## Data Availability

The data presented in this study are available on request from the corresponding authors.

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
