# Peer review of "Corticosterone-Mediated Physiological Stress Alters Liver, Kidney, and Breast Muscle Metabolomic Profiles in Chickens"

_animals, 2021, doi:10.3390/ani11113056_

Round 1

Reviewer 1 Report

  1. The manuscript lacks any information on experimental replication. This is particularly worrisome. Please revise the manuscript detailing your experimental and technical replications.
  2. What do the final experimental results mean for production? The authors can make a prospect in the conclusion.
  3. Have you tried other prescriptions besides alcohol?
  4. Check the reference list, format of the reference should be unified.

Author Response

See uploaded document.

Reviewer 2 Report

In their manuscript, Corticosterone-mediated physiological stress alters liver, kidney, and breast muscle metabolomic profiles in chickens. The authors present an interesting investigation, which requires some revisions in order to become publishable.

Specific points

  1. Why did the author choose liver, kidney and breast muscle as the research objects of metabolomic?
  2. In this study, the author selected corticosterone doses of 10 and 30 mg L -1. What was the basis of selection?
  3. It is recommended to change p-value to italic.

Author Response

See uploaded document.

Reviewer 3 Report

The manuscript reports a relevant topic for poultry science area, demonstrating the effects of physiological stress on the metabolomes of liver, kidney, and breast muscle in chickens. The subject of this study is interesting, both from a theoretical and practical point of view.  The experiment was designed well attempting to answer several questions.

Author Response

See uploaded document.

Round 2

Reviewer 1 Report

Very good answer